# SoM-1K: A Thousand-Problem Benchmark Dataset for Strength of Materials

## Abstract

Foundation models have shown remarkable capabilities in various domains, but their performance on complex, multimodal engineering problems remains largely unexplored. We introduce SoM-1K, the first large-scale multimodal benchmark dataset dedicated to evaluating foundation models on problems in the strength of materials (SoM). The dataset, which contains 1,065 annotated SoM problems, mirrors real-world engineering tasks by including both textual problem statements and schematic diagrams. Due to the limited capabilities of current foundation models in understanding complicated visual information, we propose a novel prompting strategy called Descriptions of Images (DoI), which provides rigorous expert-generated text descriptions of the visual diagrams as the context. We evaluate eight representative foundation models, including both large language models (LLMs) and vision language models (VLMs). Our results show that current foundation models struggle significantly with these engineering problems, with the best-performing model achieving only 56.6% accuracy. Interestingly, we found that LLMs, when provided with DoI, often outperform VLMs. A detailed error analysis reveals that DoI plays a crucial role in mitigating visual misinterpretation errors, suggesting that accurate text-based descriptions can be more effective than direct image input for current foundation models. This work establishes a rigorous benchmark for engineering AI and highlights a critical need for developing more robust multimodal reasoning capabilities in foundation models, particularly in scientific and engineering contexts.

## 1 Introduction

Strength of Materials (SoM) or Mechanics of Materials is a cornerstone of engineering, studying how solid objects deform and fail under loads. Solving SoM problems requires seamlessly integrating multimodal information, both textual and visual. For instance, an engineer must analyze the text describing material properties, while simultaneously interpreting diagrams that illustrate the geometry of the structure and its boundary conditions. This ability to reason across modalities is a fundamental engineering skill, yet it remains a significant challenge for AI models (Wang et al., 2025).

SoM is also a high-stakes domain, where reasoning errors can lead directly to unsafe designs and structural failures. Reliability is therefore not optional but essential: any AI system deployed in this context must meet the same rigorous standards of safety and precision expected of human engineers. These demands make SoM particularly challenging for AI, as models must not only perform accurate calculations but also correctly interpret schematics and integrate them with textual problem statements.

While foundation models have shown strong performance in text-based mathematical reasoning (Cobbe et al., 2021; Ahn et al., 2024; Seßler et al., 2024), they often struggle with specialized vision-language tasks in engineering. The main reason is that existing vision-language models (VLMs), typically trained on natural images, lack the domain-specific knowledge needed to interpret engineering schematics (Doris et al., 2024). To be useful in engineering, AI must evolve to reason from visual information with the same precision as human experts (Hao et al., 2025).

A key obstacle of developing reliable foundation models for SoM is the lack of suitable datasets. Current datasets are poorly aligned with the unique demands of engineering problem-solving (Pi-

card et al., 2023). Text-only datasets omit critical visual cues, while popular multimodal datasets focus on everyday imagery (Schuhmann et al., 2022) and exclude the specialized symbols and physical principles central to engineering. As a result, no standardized benchmark exists yet to evaluate foundation models on authentic, visually rich SoM problems. Existing evaluations largely emphasize conceptual or text-based questions (Marino et al., 2019), neglecting the multimodal reasoning required to arrive at physically grounded solutions (Bakhtin et al., 2019; Yi et al., 2020). More recently, Yue et al. (2024) introduced the MMMU benchmark to evaluate the multimodal reasoning capabilities of various models across a broad range of disciplines. To the best of the authors' knowledge, however, benchmark studies specifically designed for mechanics analysis, particularly those with detailed reasoning steps as ground truth and explicit error-type classifications, remain scarce. Developing such a benchmark is therefore essential to systematically assess models' capabilities and advance their reliable use in engineering practice.

To bridge this gap, we introduce **SoM-1K**, the first domain-specific multimodal benchmark that integrates text, equations, and engineering diagrams to reflect the authentic reasoning demands of engineering problems. Unlike prior datasets relying primarily on texts (Hendrycks et al., 2021; Wang et al., 2019), SoM-1K captures the multimodal nature of real engineering practice and establishes a standardized platform for rigorous evaluation. Our contributions are threefold: (1) we present the first large-scale multimodal benchmark tailored to mechanics problems; (2) we systematically evaluate leading foundation models, revealing their current limitations in visual-textual reasoning; and (3) we propose and validate the use of Description of the Image (DoI) as an effective prompting strategy to improve the perception capacity of current foundation models. Together, these contributions not only fill a critical gap in AI evaluation resources but also provide actionable insights for developing the next generation of AI systems capable of reliable engineering reasoning.

## 2 RELATED WORK

**Foundation Models in STEM (Science, Technology, Engineering, and Mathematics).** The use of Large Language Models (LLMs) in STEM has grown rapidly. Initially, models like Minerva (Lewkowycz et al., 2022) and PaLM (Chowdhery et al., 2022) excelled at solving complex math and physics problems by using techniques like chain-of-thought (CoT) prompting (Wei et al., 2023). This success has expanded into various engineering disciplines, where LLMs assist with design, simulations (Liu et al., 2024), and inverse problems, often by integrating with external tools (Niketan et al., 2025). For instance, LLMs are being applied in bridge engineering to interpret and process the vast amount of unstructured data found in inspection reports, transforming it into structured, actionable insights for decision support (Kumar & Agrawal, 2025). The development of VLMs has also been crucial, allowing models to interpret diagrams and schematics (Picard et al., 2024), a core part of engineering education. These VLMs are now used in educational settings to provide interactive, step-by-step guidance by analyzing visual inputs (Bewersdorff et al., 2025; Scarlatos et al., 2025).

**Benchmarking in Engineering Domains.** Existing benchmarks in engineering domains have highlighted the challenges faced by AI models in interpreting and reasoning over technical diagrams and textual information. For instance, the DesignQA benchmark evaluates VLMs on tasks involving engineering documentation, CAD images, and textual design requirements, revealing significant gaps in model performance when both visual and textual information are required (Doris et al., 2024). Similarly, the EEE-Bench benchmark assesses VLMs on practical engineering tasks in electrical and electronics engineering, demonstrating that current models often struggle with complex visual and textual integration, achieving average performance ranging from 19.48% to 46.78% (Li et al., 2025b). These studies underscore the necessity for benchmarks that rigorously evaluate AI models' abilities to handle multimodal engineering problems, including the integration of schematic diagrams and textual descriptions.

**AI Assistance in Mechanics of Materials.** In the field of mechanics of materials, several projects have explored the use of AI and LLMs (Tian & Zhang, 2024; Buehler, 2023; Ni & Buehler, 2023; Liu et al., 2025). For instance, the AutoGen (Tian & Zhang, 2024) aimed to presents a framework where multiple LLM-based agents collaborate to solve mechanics problems using the Finite Element Method. The MechAgent (Ni & Buehler, 2023) introduced a novel multi-agent paradigm where a

team of AI agents with specialized roles collaboratively automates the process of solving complex mechanics tasks.

To the best of our knowledge, however, no multimodal benchmark study has yet evaluated the reasoning capabilities of foundation models in solving mechanics problems.

## 3 THE SoM-1K DATASET

### 3.1 BACKGROUND IN SoM

SoM is a fundamental branch of engineering that studies how solid objects respond to external forces, such as tension, compression, torsion, and bending. Problems in this domain typically focus on analyzing why materials fail, a fundamental concern underlying nearly all engineered systems, from bridges and aircraft to robots and microchips. For this reason, SoM is a core subject in civil, mechanical, aerospace, and materials engineering curricula worldwide, and accurate problem-solving in this domain underpins real-world engineering design and decision-making. Hence, SoM provides an ideal domain for evaluating foundation models, as it requires the integration of physical principles, mathematical formulations, and the logical application of boundary conditions, paralleling the forms of reasoning demanded in complex coding and scientific problem-solving.

### 3.2 SCOPE OF THE DATASET

Our benchmark dataset, **SoM-1K**, is designed to evaluate AI models on authentic mechanics problems. It includes the three fundamental problem types: axial loading (bars), torsion (shafts), and bending (beams and frames) (Hibbeler, 2012). SoM-1K spans a wide range of calculation tasks, including computation of internal forces, stresses, strains, and deformations, diagram construction, and design-oriented optimizations.

Problems were carefully selected from widely-used university textbooks (Sun et al., 2009; Huang, 2009; Dai, 2015; Ma, 2011; Hibbeler, 2012; Wight et al., 2011) and advanced mechanics competitions, ensuring a hierarchical dataset encompasses both routine exercises and more challenging tasks. All source materials were consolidated into PDF format, with textbooks scanned from physical copies and competition problems obtained from official exam websites (Chinese Society of Theoretical and Applied Mechanics & Zhou Peiyuan Foundation, 2025).

In total, SoM-1K comprises 1,065 annotated problems, summarized in Table 1, categorized into five groups based on structural components and loading conditions: (1) Axial loading (bars), (2) Torsion (shafts), (3) Bending-I (beams), (4) Bending-II (frames), and (5) Integrated tasks. Integrated problems, sourced from mechanics competitions, require multi-concept reasoning, combining static analysis with dynamic concepts such as vibration, impact, and rigid-body motion. Example problems from each category are provided in Figure 7 (Appendix A).

Table 1: Statistics of dataset composition in SoM-1K.

| Category | Quantity | Proportion |
|---|---|---|
| **Classified by deformation modes** | | |
| Axial loading (bars) | 201 | 18.87% |
| Torsion (shafts) | 137 | 12.86% |
| Bending-I (beams) | 630 | 59.15% |
| Bending-II (frames) | 54 | 5.07% |
| Integrated tasks | 43 | 4.04% |
| **Overall** | **1065** | 100% |
| **Classified by statical indeterminacy** | | |
| Statically determinate (easy) | 917 | 86.10% |
| Statically indeterminate (hard) | 148 | 13.90% |

### 3.3 COMPONENTS OF THE DATASET

An illustrative example of the dataset structure is shown in Figure 1. Each problem consists of four standardized components:

**(1) Problem Statement (PS):** A concise textual description of the problem that specifies the given information and the quantity or outcome to be determined.

**(2) Schematic Diagram (Image, I):** A graphical representation of the structure or an object, provided in image format. Throughout this work, the term *Image* refers to such schematic diagrams.

**(3) Description of the Image (DoI):** Expert-validated text describing schematics (e.g., geometry,

boundary conditions), providing a precise representation of visual information for evaluating model performance.

**(4) Ground Truth (GT):** The correct solution to the problem, including equations, reasoning steps, and final answers.

---

**Problem statement (PS)**

A simple beam with an overhang supports a uniform load of intensity $q$ on span $AB$ and a concentrated load $P$ at end $C$ of the overhang. Determine the deflection $\delta_C$ and angle of rotation $\theta_C$ at point $C$. (Use the modified form of Castigliano's theorem.)

**Image (I)**

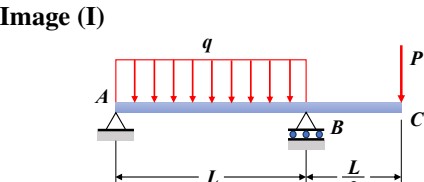

**Description of the image (DoI)**

This diagram shows a simply supported beam structure with an overhang.

*Structure*: Beam AC is supported by two supports: the left end $A$ is a hinge support, and point $B$ is a roller support. Segment $AB$ is the main span of the beam, with a length of $L$. Segment $BC$ is the overhanging part extending beyond support $B$, with a length of $L/2$.

*Loading*: The beam is subjected to two combined loads: a uniform load with an intensity of $q$ acts on the main span $AB$ downward; a concentrated load with a magnitude of $P$ acts at point $C$, the end of the overhang downward.

**Ground Truth (GT)**

Deflection $\delta_C$ at the end of the overhang. Since the load $P$ corresponds to this deflection, we do not need to supply a fictitious load. Instead, we can begin immediately to find the bending moments throughout the length of the beam. [...]

After carrying out the integrations and combining terms, we obtain

$$\theta_C = \frac{7PL^2}{24EI} - \frac{qL^3}{24EI}.$$

---

Figure 1: An illustrative example of the dataset structure : problem statement (PS), image (I), description of the image (DoI), and ground truth (GT).

Our workflow began with scanned files of textbooks and problem sets, followed by a two-step pre-processing pipeline. First, schematic diagrams were manually extracted and stored as PNG images (size ranging from 195x104 to 850x688), uploaded to the cloud, and fed into models via URL links. Second, textual content, including PS and GT, was extracted using Doubao (ByteDance, 2025) Optical Character Recognition (OCR). If the GT includes internal force diagrams (Hibbeler, 2012) or other elements that cannot be extracted via OCR, the annotation team manually supplement the description of these diagrams. The extracted text was then carefully reviewed and manually refined to correct OCR errors, ensuring accurate and high-quality representations. The annotation team includes experienced researchers and educators in structural engineering and mechanics of materials, including a PhD candidate, two lecturers, and four teaching assistants.

During preliminary testing, we observed that foundation models struggled to process LaTeX-formatted expressions in batch inference. To mitigate this, we employed the DeepSeek-V3-0324 API (SiliconCloud, 2025) to convert all LaTeX equations into natural-language descriptions, thereby providing consistent textual representations for model inputs.

### 3.4 DoI Annotation Process

The DoI is derived from the PNG schematics (see Figure 2). Each image is first processed by the Doubao (ByteDance, 2025) VLM, which generates an initial textual description capturing key aspects of the structural diagrams, including geometry, loading conditions, and boundary conditions. These auto-generated descriptions are then carefully reviewed and refined by the annotation team to correct errors and incorporate missing information critical for problem-solving.

It is important to note that our DoI is fundamentally different from a typical CoT prompt (Wei et al., 2023). The DoI is designed to only describe the information visually present in the image and does not provide any additional insights or step-by-step reasoning to help the model solve the problem. This clear distinction allows us to isolate and measure the specific impact of descriptive image information on the model's performance.

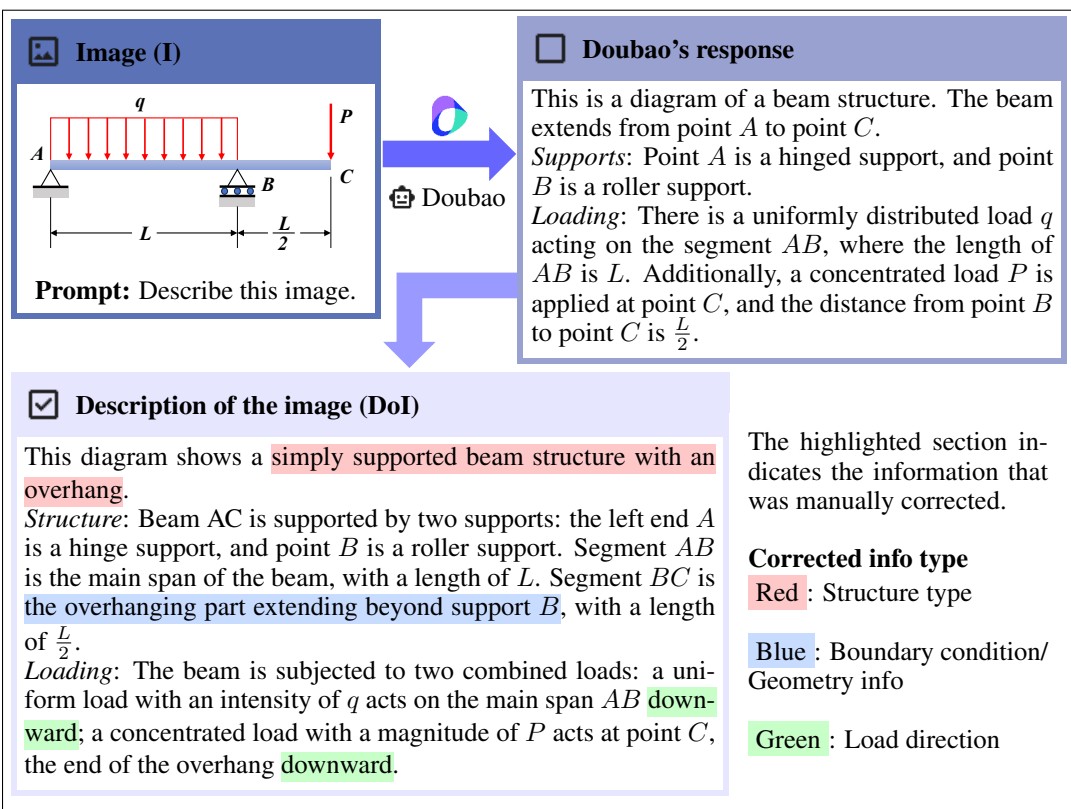

Figure 2: Workflow illustrating the process from the input image to the DoI: an image is first processed by the Doubao VLM, which generates an initial description of the image. This response is then carefully reviewed and corrected by human experts to produce the final DoI. (*For improved readability of this colored figure, please refer to the digital version of the paper.*)

## 4 EVALUATION

### 4.1 MODELS SELECTED

We evaluate eight representative foundation models on our collected dataset. To ensure a diverse representation of the current landscape, we include both closed-source models (e.g., GPT-4o (OpenAI, 2024), Qwen-plus (Alibaba Cloud, 2025a), Qwen-VL (Alibaba Cloud, 2025b), GPT-3.5 (OpenAI, 2023) and Doubao (ByteDance, 2025)) and leading open-source models (e.g., Llama-70B (Meta AI, 2024), GPT-oss-120b (OpenAI, 2025) and DeepSeek-R1 (DeepSeek, 2025)).

Among them, LLMs include GPT-oss-120b, Qwen-plus, DeepSeek-R1, GPT-3.5, and Llama-70B, while VLMs include Doubao, Qwen-VL, and GPT-4o. This selection provides a broad range of training architectures and accessibility. A full list is provided in Table 2 (Appendix C).

### 4.2 EVALUATION PROTOCOL

**Prompting Strategy.** To comprehensively evaluate the performance of different foundation models, we designed three prompting strategies, **(1) PS+I; (2) PS+I+DoI; (3) PS+DoI**, as illustrated in Figure 3. VLMs were evaluated under all three prompting strategies according to their multi-

modal capabilities. In contrast, LLMs were evaluated only under the **PS+DoI** setting, reflecting their text-only input constraints. This design enables a systematic comparison across modalities: (i) whether textualizing diagrams improves reasoning, (ii) whether incorporating schematics enhances performance, and (iii) how visual versus textual representations differentially affect outcomes. Additionally, we did not impose any token length restrictions or modify parameters such as temperature, and all settings were used with their default values, ensuring that the model could complete the full reasoning process and output complete results.

**Majority Voting.** To mitigate random variability in the model's output, we adopt a robust evaluation protocol. For each problem and prompt strategy, we generated five independent responses. The final answer was then determined using a majority-vote mechanism, which was implemented using the DeepSeek-V3-0324 (SiliconCloud, 2025) API. This helps to ensure the reliability of our results.

**Human Evaluation.** The mainstream approach in recent work for evaluation is to use LLMs-as-Judge (Li et al., 2025a). Our pilot study tested on the Qwen-plus (Alibaba Cloud, 2025a) API on 200 sample problems showed that automated grading achieved an 83% agreement with expert judgments. Despite this encouraging result, we ultimately chose to rely on manual evaluation by human experts for the full dataset, given its manageable size and the importance of performing detailed error analyses that automated systems currently cannot provide with sufficient reliability.

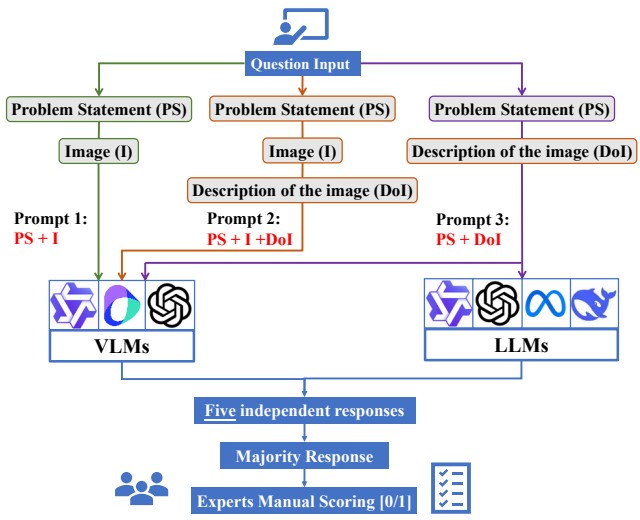

Figure 3: Each problem is tested under 14 model–prompt settings (three strategies × three VLMs, plus PS+DoI × five LLMs). For each setting, five responses are generated, majority-voted, and scored by experts with binary labels (1 = correct, 0 = incorrect)

For the human evaluation, the DoI annotation expert team collectively examined each model-generated response across all 1,065 problems. The evaluation proceeded in two stages: first, verifying whether the reasoning process followed a logically valid sequence of steps, and second, checking whether the final answer was correct. A response was awarded a score of 1 only if both criteria were satisfied; otherwise, it received a score of 0. Such binary grading method simplified the process and also reduced the likelihood for inconsistencies. In cases where decisions were difficult or the outcome was ambiguous, the evaluators engaged in discussion until consensus was reached. This process not only provided high-quality ground truth labels but also enabled the identification of systematic error patterns. The overall model performance is reported using Accuracy.

## 4.3 RESULTS

We report results for all compared foundation models with different prompting strategies for our proposed benchmark dataset SoM-1K in Figure 4, Table 3, and Figure 8 (see Appendix C for more details). We have the following observations: (1) The best-performing model, Qwen-plus achieved an accuracy of 56.6% using the PS+DoI prompt strategy, while GPT-3.5 scored the lowest at 1.0%. The observed low ratings highlight the significant challenges that current foundation models face when addressing engineering problems. (2) The top-performing models were **Qwen-plus (56.6%), Deepseek-R1 (52.4%), and Doubao (48.5%)**, all of which achieved their best results using the **PS+DoI prompting strategy**. It is also interesting to note that, for VLMs like Doubao and GPT-4o, including an image in the prompt (PS+I or PS+DoI+I) barely improves performance compared to the text-only PS+DoI prompt. (3) With the exception of Doubao, text-only reasoning models (**Qwen-plus, DeepSeek-R1, and GPT-oss-120b**) generally outperformed the VLMs (**Qwen-VL and GPT-**

**4o**) evaluated in this work. (4) Among open-source models, larger LLMs generally perform better: DeepSeek-R1 (**671B**) achieves **52.4%** accuracy, GPT-oss (**120B**) **39.0%**, and Llama (**70B**) only **9.1%**.

Additionally, the performance of each model across different categories of problems is presented in Figure 9 (Appendix C). A closer look at the category-specific performance charts reveals a significant variation in model capabilities across different engineering problem types. While simpler tasks like Torsion (shafts) see higher accuracy (e.g., **75.9%** for Deepseek and Qwen-plus), more complex problems such as Bending-II (frames) and Integrated tasks remain extremely challenging, with the highest accuracy for frames being just **16.7%** for Qwen-plus and only **11.6%** for Deepseek in Integrated tasks. In fact, over half the models scored below **3.7%** in these categories, highlighting the difficulty of multi-step reasoning and the integration of physical principles. Additionally, the impact of the prompting strategy is evident, as **PS+DoI** consistently outperformed **PS+I** in most tasks.

These findings indicate (1) VLMs' limited capabilities in interpreting and integrating domain-specific information from schematic diagrams, suggesting the need for further advancement; (2) the relative effectiveness of well-structured textual information for aiding foundation models' complex problem-solving; and (3) the significant challenge that multi-step reasoning and the integration of advanced physical principles pose for current models, highlighting the need for more sophisticated approaches in solving advanced engineering problems.

To establish a meaningful performance baseline, we conducted a human evaluation with four teaching assistants from the field of structural engineering, who were tasked with solving a randomly selected set of 100 SoM problems, with each assistant solving 25 problems, within two days, referring only to the textbooks (formulas, general knowledge, etc.). They achieved 95% accuracy, whereas Qwen-plus, with a best accuracy of 56.6%, remains well below human-level performance.

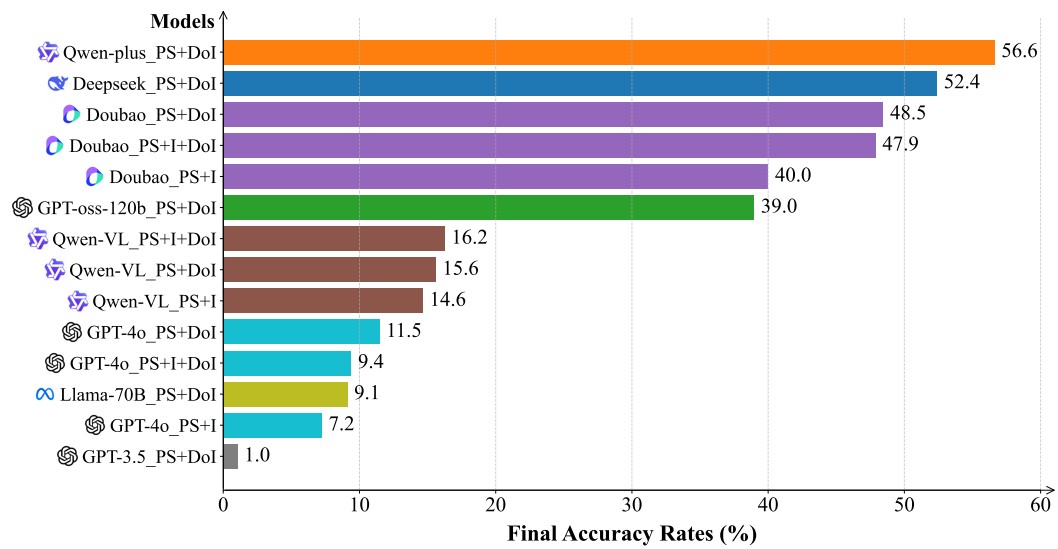

Figure 4: Accuracy for each evaluated model with different prompting strategies.

## 5 DISCUSSIONS

### 5.1 WHAT TYPES OF ERRORS CAN FOUNDATION MODELS MAKE IN SOM-1K?

To gain deeper insights into the capabilities of foundation models in solving mechanics problems, we engaged human experts to examine problems that none of the models were able to solve. Following a thorough manual review, we developed a comprehensive error taxonomy, illustrated with examples in Figure 5. Specifically, we classify the errors into four distinct categories:

- **Type K (Knowledge-based Error)**: Failing to apply correct domain knowledge, e.g., misjudging internal loads in a structure.

- **Type C (Calculation Error)**: Correct formulas used but numerical results are wrong.
- **Type E (Extraction Error)**: Failing to interpret or extract information from the prompts, producing misaligned answers.
- **Type O (Other Error)**: Incomplete solutions or responses that entirely miss the problem.

Knowledge-based (K) and calculation (C) errors typically indicate that the model has grasped the problem but faltered in recalling domain knowledge or performing arithmetic mistakes that could potentially be mitigated through external tools, e.g., retrieval augmentation and computational simulations. In contrast, extraction (E) and other (O) errors demonstrate that the model either failed to interpret the problem correctly or was unable to construct a coherent solution pathway. Such failures point to a breakdown in comprehension and reasoning, directly constraining the model's ability to engage with mechanics problems.

| K (Knowledge-based Error) | C (Calculation Error) |
| --- | --- |
| **Response:**
• Taking moments about point D:
$\boxed{V_B \cdot a - P_2 \cdot b = 0}$ | **Response:**
• It is calculated that:
$$d \geq \left(\frac{16000}{30\pi}\right)^{1/3} \boxed{\approx 2.123 \text{ cm}}$$ |
| **Comment:**
The sign for $P_2$ should be positive and the correct equation should be: $\boxed{V_B \cdot a + P_2 \cdot b = 0}$ | **Comment:**
The result is miscalculated and the correct answer should be:
$$d \geq \left(\frac{16}{30000\pi}\right)^{1/3} \boxed{\approx 55.4 \text{ mm}}$$ |
| **E (Extraction Error)** | **O (Other Error)** |
| **PS:** Draw the moment diagrams for the beam using the method of superposition.
**Response:**
• 2. Segment $5 < x \leq 20$ (span A to B)
• Bending moment formula: $M = -525 + 15s - \frac{s^3}{18}$
• Parameter definition: $s = x - 5$ | **Response:**
• 1. Primary Structure:[. . . ]
• 2. Redundant Structure:[. . . ]
• Feel free to ask if you have any specific questions or need further clarification on this process.
End of the response. |
| **Comment:**
The method of superposition is not used. | **Comment:**
The final solution is not provided. |

Figure 5: Description of the four error types.

To analyze the types of errors made by foundation models, we randomly selected 100 problems that all models failed to solve. For each problem, the annotation team manually reviewed the majority response from each model and categorized it into one of four predefined error types (K, C, E, O). The reviewers first examined whether the response contained a complete problem-solving process. If the model either failed to provide a full solution or made no attempt to solve the problem, the response was labeled as Type O. Otherwise, the reviewers carefully traced the solution from the beginning, identified the earliest mistake, and assigned it to Type E, C, or K.

## 5.2 WHAT ERRORS DO DIFFERENT FOUNDATION MODELS MAKE?

The distribution of error types was then computed as the percentage of responses in each category out of the 100 problems. These proportions are reported in Figure 6 as **Percentage**.

As shown in Figure 6, while all models failed on these 100 problems, their error distributions differed markedly. In particular, Figure 6(a–c) show that GPT-3.5, GPT-4o_PS+I, Qwen-VL_PS+I, and Llama-70B exhibited a high proportion of Type O and Type E errors, with more than 39% of their failures reflecting a fundamental misunderstanding of the problem. This pattern is consistent with their lower overall accuracy in Figure 4. In contrast, Qwen-plus and DeepSeek-R1 demonstrated substantially fewer critical errors (Type O+E error rates of 7% or less), which aligns with their stronger overall performance. These results suggest that the latter models possess a more reliable grasp of the underlying problem-solving logic.

Notably, under **PS+I**, **Qwen-VL** and **Doubao** show high Type E errors (34% and 19%) versus Type O errors (5% and 1%), reflecting difficulties in extracting visual information. In contrast, **GPT-4o_PS+I**, **GPT-3.5**, and **Llama-70B** exhibit the opposite trend, with Type O errors dominating within the O+E category, indicating challenges in reaching final solutions in this domain.

Our earlier results demonstrated that incorporating DoI enhances the performance of VLMs. To better understand the mechanism driving this improvement, we compared Doubao, Qwen-VL, and GPT-4o under three prompting strategies. The results, summarized in Figure 6b, show that prompting with DoI markedly reduces the frequency of Type E errors. For instance, Doubao's Type E error rate dropped from 19% under PS+I to 3% under PS+DoI and 5% under PS+I+DoI. Similarly, Qwen-VL's Type E error rate decreased from 34% (PS+I) to 6% (PS+DoI) and 5% (PS+I+DoI), while GPT-4o's rate fell from 10% (PS+I) to 2.0% (PS+DoI) and 3.0% (PS+I+DoI). These substantial reductions highlight DoI's effectiveness in mitigating misinterpretations of visual information, thereby supporting more accurate problem-solving.

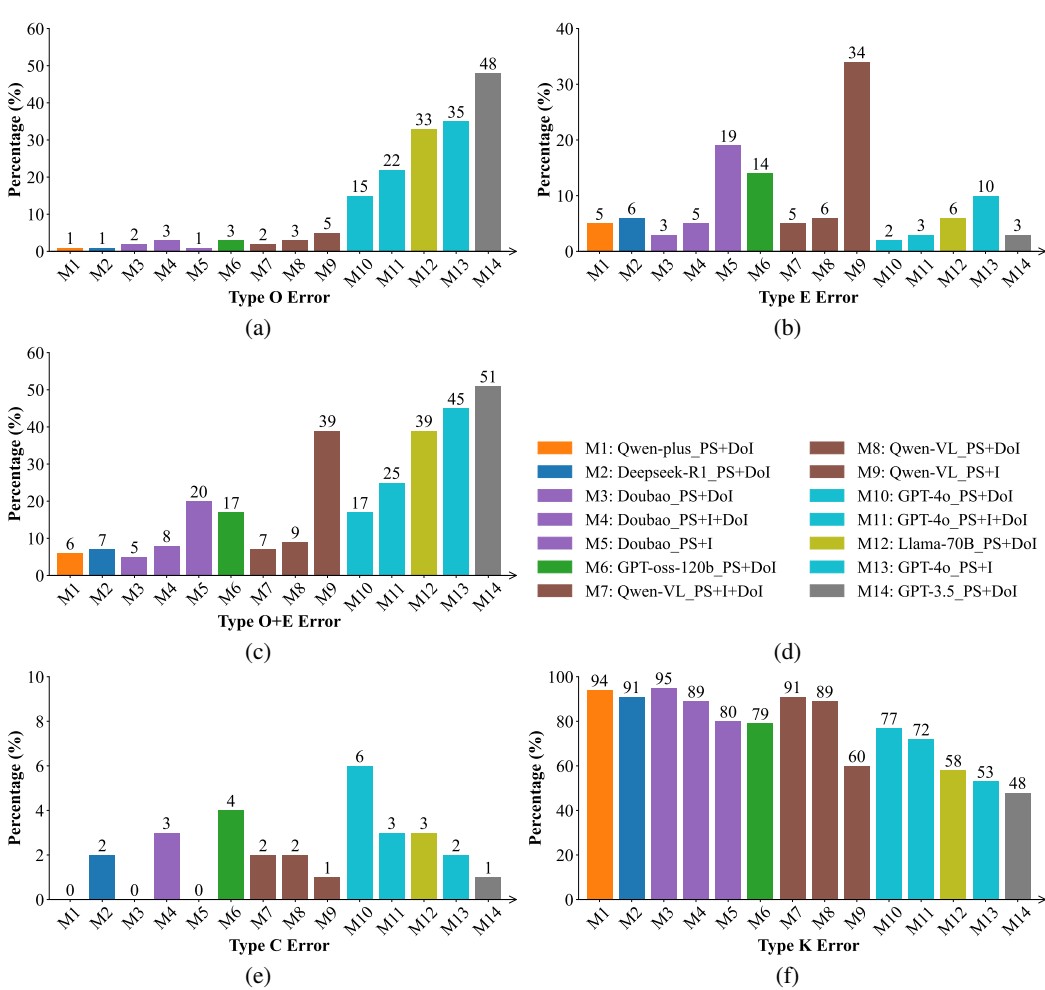

Figure 6: The percentage of error types of each model among 100 questions that all models fail to solve. (a) Type O, (b) Type E, (c) Type O+E, (d) legend, (e) Type C, (f) Type K.

As illustrated in Figure 6e, arithmetic errors (Type C) still occur, demonstrating that models may miscompute even when the correct formula is used. Among all four error types, Type K errors are the most frequent (Figure 6f), reflecting gaps in engineering knowledge that could be mitigated via supervised fine-tuning on domain-specific data.

### 5.3 CAN FOUNDATION MODELS PROVIDE BETTER SOLUTIONS THAN TEXTBOOKS?

Another interesting case study is that foundation models can sometimes generate better answers. As shown in Figure 10 (Appendix C), the correct solutions generated by Qwen-plus is more detailed and pedagogically structured than the textbook solution. This suggests that foundation models have a strong potential to be used in educational applications, providing richer and more comprehensive explanations for students.

## 6 CONCLUSION

This study introduced SoM-1K, a novel multimodal benchmark for evaluating the problem-solving abilities of foundation models in strength of materials. Unlike previous text-only benchmarks, SoM-1K uses a combination of text and schematic diagrams to provide a more realistic and rigorous evaluation. Our findings reveal that even the most advanced LLMs and VLMs struggle with these complex, domain-specific engineering problems, showing significant limitations in their reasoning capabilities. We also demonstrated that using DoI as a prompting strategy dramatically improves performance by reducing misinterpretation errors. This suggested that, compared with visual data, well-structured textual input may be semantically richer and could serve as a more reliable foundation for complex reasoning.

Several limitations should be addressed in future work. One key area is the impact of image rendering and diagram parsing on the reasoning performance of VLMs. Additionally, our current binary correctness score does not capture partial reasoning, such as correct free-body setups with numerical errors, and a more granular step-by-step or partial-credit evaluation would provide deeper insights into model reasoning. Finally, due to budget constraints, we were unable to include the latest state-of-the-art models like Gemini 2.5 Pro or o3, which would likely strengthen our analysis. Including these models in future evaluations could further validate our findings and provide stronger evidence for our conclusions.

Additionally, the scalability of the DoI annotation process is another challenge, and we are developing a system for automated DoI generation through VLM fine-tuning for a larger dataset. Future research should focus on expanding the scope of multimodal benchmarks beyond the current limitations of SoM-1K to include more advanced engineering domains like structural dynamics, plasticity, and nonlinear mechanics. SoM-1K could also be leveraged for practical applications, such as model fine-tuning for specific engineering tasks and AI tutoring to help students improve problem-solving skills and understand complex concepts. The persistent challenges observed in diagram-based reasoning and calculation errors highlight a critical need for future models to enhance their **multimodal reasoning capabilities** and to integrate more effectively with specialized tools. This will enable them not only to solve complex problems but also to reliably generate accurate scientific diagrams, such as internal force diagrams or deformation shapes, which remains a significant hurdle for current foundation models.

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

# APPENDIX

## A  DATASET EXAMPLES

To illustrate the diversity of SoM-1K, Figure 7 shows one representative problem from each dataset category.

## B  GUIDELINE FOR GENERATING DoI ANNOTATIONS

The Description of Image (DoI) provides a clear description of the visual elements in engineering schematics. To generate a DoI, follow these steps:

1. Start with the original PNG schematic diagram and process it through the Doubao VLM, which will generate an initial description capturing the geometry (shapes and dimensions of components), loading (applied forces, moments, directions and points of application), and boundary conditions (supports);

2. Review and refine the description by correcting errors and adding any missing details from the diagram, such as hidden supports or load directions;

3. Perform a final check: the DoI should strictly describe what is visible in the diagram without any assumptions or reasoning, do not infer reaction forces or deflections that are not shown. Use clear, technical language, organizing the description into sections (Geometry, Loading, Boundary Conditions).

### ✏️ Axial loading (bars)

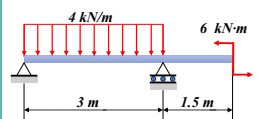

The truss shown in the figure is subjected to a vertical load $F$ at node $C$, where rod 3 is a rigid rod, and the length and tensile and compressive stiffness $EA$ of rods 1 and 2 are the same. Find the internal force of each rod.

### 🔄 Torsion (shafts)

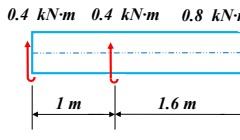

A circular shaft with a diameter of $55\,$mm is loaded as shown in the figure, and its allowable shear stress is $[\tau] = 30\,$MPa. Try to draw the torque diagram of the shaft and check its torsional strength.

### 🔁 Bending-I (beams)

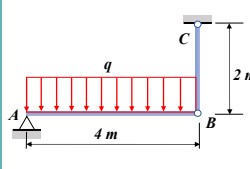

Draw the shear and moment diagrams for the beam.

### 🔲 Bending-II (frames)

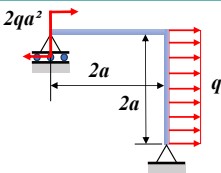

Draw the axial force, shear force and bending moment diagrams of the rigid frame shown.

### ▱ Statically determinate

Both beam $AB$ and rod $CB$ have circular cross-sections and are made of the same material. The modulus of elasticity is $E = 200\,$GPa, the allowable stress is $[\sigma] = 160\,$MPa, and the diameter of rod $CB$ is $d = 20\,$mm. Under the load shown in the figure, the measured axial elongation of rod $CB$ is $\Delta l_{CB} = 0.5\,$mm. Find the value of the load $q$ and the safe diameter of beam $AB$.

### ⊞ Statically indeterminate

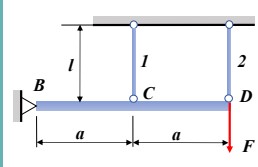

In the structure shown, the beam $BD$ is a rigid beam, and rods 1 and 2 are made of the same material. The cross-sectional areas are both $A = 300\,$mm$^2$, the allowable stress is $[\sigma] = 160\,$MPa, and the vertical load at point $D$ is $F = 50\,$kN. Check the strength of rods 1 and 2.

### 👥 Integrated tasks

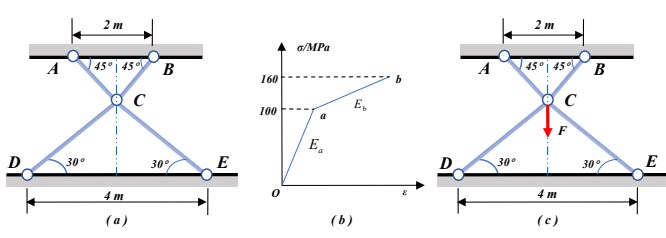

A planar truss structure consists of four circular-section straight bars made of the same material. Among them, bar AC and bar BC have the same length and a diameter of $d_1 = 20$mm, while bar CD and bar CE have the same length and a diameter of $d_2 = 40$mm. The design dimensions are shown in Figure (a). The stress-strain curve of each bar's material is shown in Figure (b) (piecewise linear), and the elastic moduli of segments Oa and ab are $E_a = 200$GPa and $E_b = 50$GPa respectively. During assembly, it is found that both bar AC and bar BC are shorter than the design dimension by 0.3mm. Find the internal forces of each bar after assembly is completed;

After assembly is completed, apply a vertically downward force $F = 90$kN at point C, as shown in Figure (c), find the internal forces of each bar.

Figure 7: Illustrative examples of one representative problem from each category in the SoM-1K dataset.

## C  ADDITIONAL TABLES AND FIGURES

Table 2: Overview of evaluated foundation models, including source availability, release year, modality and size.

| Model | Full Name | Open Source | Team | Time(M/Y) | Modality | Size |
|---|---|---|---|---|---|---|
| Doubao (ByteDance, 2025) | Doubao-1.5-thinking-vision-pro-250428 | Closed | ByteDance | 04/2025 | VLM | N/A |
| Qwen-plus (Alibaba Cloud, 2025a) | Qwen-plus-2025-07-28 | Closed | Alibaba Cloud | 07/2025 | LLM | N/A |
| Qwen-VL (Alibaba Cloud, 2025b) | Qwen-VL-Max-2025-04-08 | Closed | Alibaba Cloud | 04/2025 | VLM | N/A |
| Deepseek-R1 (DeepSeek, 2025) | Deepseek-R1-0528 | Open | DeepSeek | 05/2025 | LLM | 671B |
| GPT-oss-120b (OpenAI, 2025) | GPT-oss-120b | Open | OpenAI | 08 /2025 | LLM | 120B |
| GPT-4o (OpenAI, 2024) | GPT-4o-2024-08-06 | Closed | OpenAI | 08/2024 | VLM | N/A |
| GPT-3.5 (OpenAI, 2023) | GPT-3.5-turbo-0125 | Closed | OpenAI | 11/2023 | LLM | N/A |
| Llama-70B (Meta AI, 2024) | Llama-3.3-70B-instruct | Open | Meta AI | 12/2024 | LLM | 70B |

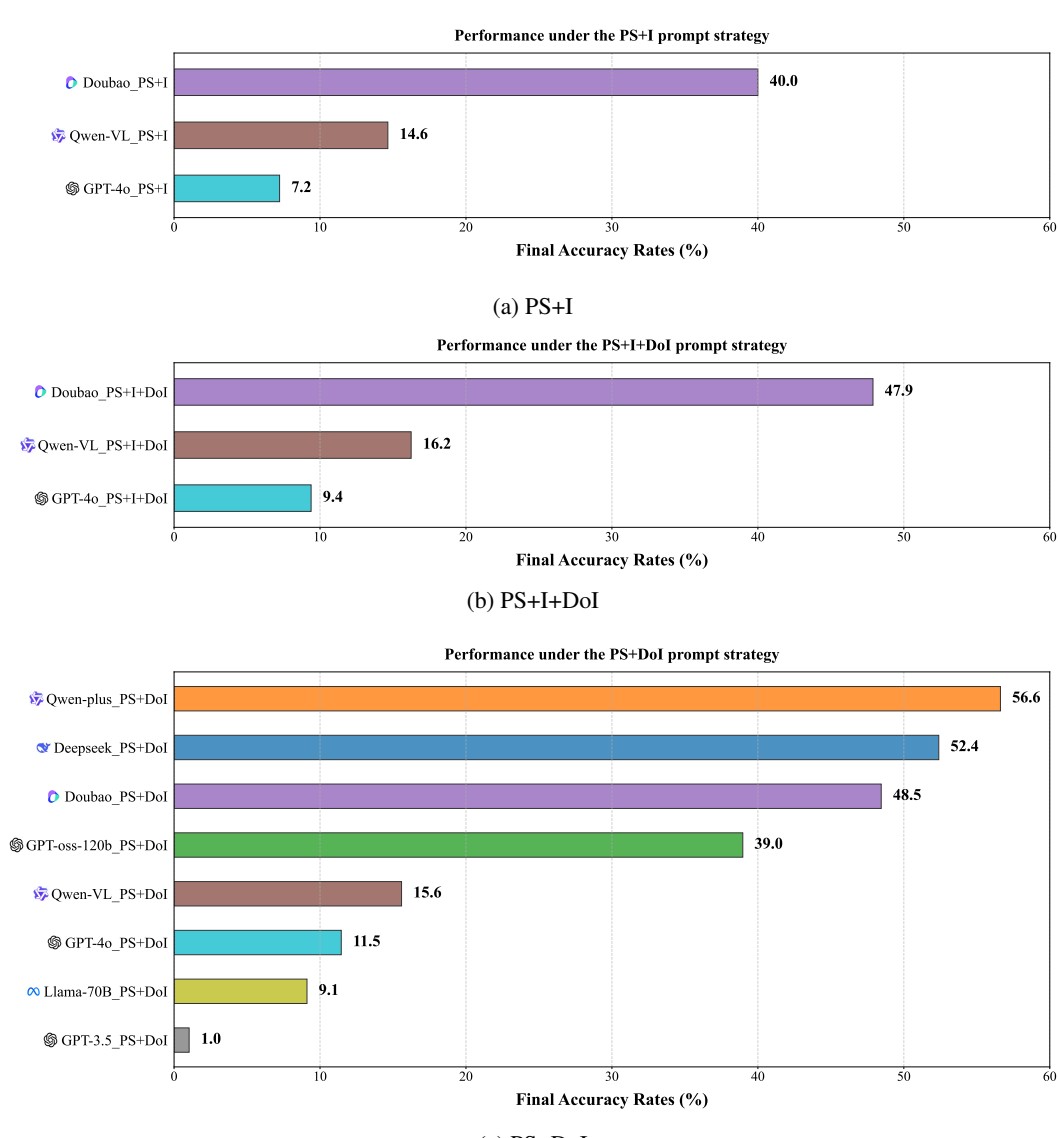

(a) PS+I

(b) PS+I+DoI

(c) PS+DoI

Figure 8: Performance of different models under three prompting strategies: (a) PS+I, (b) PS+I+DoI, and (c) PS+DoI.

Table 3: Performance of evaluated foundation models under three prompting strategies.

| Model | Open Source | PS+I | PS+I+DoI | PS+DoI |
|---|---|---|---|---|
| Qwen-plus | Closed | – | – | 56.6 |
| Deepseek | Open | – | – | 52.4 |
| Doubao | Closed | 40.0 | 47.9 | 48.5 |
| GPT-oss-120b | Open | – | – | 39.0 |
| Qwen-VL | Closed | 14.6 | 16.2 | 15.6 |
| GPT-4o | Closed | 7.2 | 9.4 | 11.5 |
| Llama-70B | Open | – | – | 9.1 |
| GPT-3.5 | Closed | – | – | 1.0 |

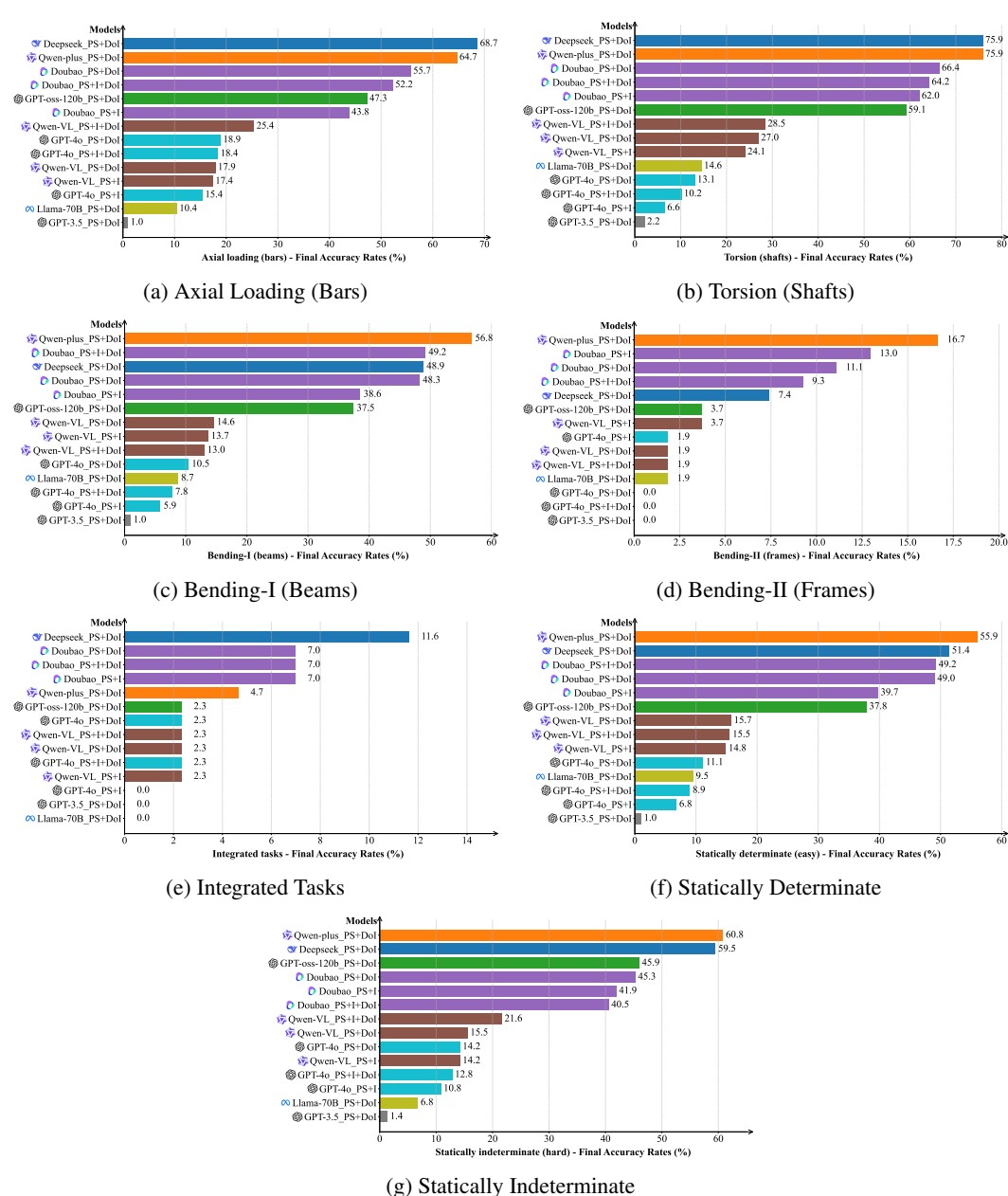

(a) Axial Loading (Bars)

(b) Torsion (Shafts)

(c) Bending-I (Beams)

(d) Bending-II (Frames)

(e) Integrated Tasks

(f) Statically Determinate

(g) Statically Indeterminate

Figure 9: Performance of different models on different categories of problems. (a) Axial Loading, (b) Torsion, (c) Bending-I, (d) Bending-II, (e) Integrated Tasks, (f) Statically Determinate, (g) Statically Indeterminate.

**Problem statement (PS)**

A simply supported beam with a circular tube cross-section is loaded as shown in the figure. It is known that $d/D = 7/10$, and the allowable stress of the material $[\sigma] = 160$ MPa. Try to find the required outer diameter $D$ and inner diameter $d$.

**Image (I)**

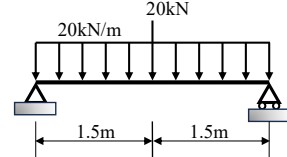

**Ground Truth (GT)**

The maximum bending moment of the circular tube section:

$$M_{\text{max}} = \frac{pl}{4} + \frac{ql^2}{8} = 37.5 \text{ kN·m}$$

From the maximum stress $\sigma_{\text{max}} = \frac{M_{\text{max}}}{W} \leq [\sigma]$, we get:

$$D^3 = \frac{32M_{\text{max}}}{\pi(1 - \alpha^4)[\sigma]}$$

Solving these equations, we obtain:

$$D = 147 \text{ mm}, \quad d = 103 \text{ mm}$$

**Qwen-plus Response (better than GT)**

1. Force Analysis of the Beam
- Support Reactions
Due to the symmetric load, the support reactions are $R_A = R_B = \frac{80}{2} = 40$ kN.
- Maximum Bending Moment
The maximum bending moment occurs at the midpoint of the beam, caused by the combined action of the uniformly distributed load and the concentrated force.

2. Bending Stress Condition
- Bending Stress Formula

$$\sigma_{\text{max}} = \frac{M_{\text{max}}}{W_z} \leq [\sigma]$$

Here, the allowable stress $[\sigma] = 160$ MPa $= 160$ N/mm$^2$. Therefore:

$$W_z \geq \frac{M_{\text{max}}}{[\sigma]} = \frac{37.5 \times 10^6}{160} = 234,375 \text{ mm}^3$$

3. Calculation of Section Modulus for Hollow Circular Section
- Section Modulus for Bending

$$W_z = \frac{\pi(D^4 - d^4)}{32D}$$

Given $\frac{d}{D} = \frac{7}{10} \implies d = 0.7D$, substituting this into the formula gives:

$$W_z = \frac{\pi D^4(1 - 0.7^4)}{32D} = \frac{\pi D^3 \times 0.7599}{32} \approx 0.0746D^3$$

- Solving for $D$
Let $0.0746D^3 \geq 234,375$, then:

$$D^3 \geq \frac{234,375}{0.0746} \approx 3,141,593 \implies D \geq \sqrt[3]{3,141,593} \approx 146.5 \text{ mm}$$

- Solving for $d$

$$d = 0.7D = 0.7 \times 146.5 \approx 102.55 \text{ mm}$$

Figure 10: Comparison between solutions generated by foundation models and textbook ground truth.

## D  Statement on the Use of Large Language Models (LLMs)

During the preparation of this manuscript, we employed LLMs for grammar correction and language refinement to enhance clarity and readability. All generated text was thoroughly reviewed, edited, and, when necessary, substantially revised by the authors, who retain full responsibility for the final content.

## E  Supplementary Materials

We provide the problem statements (PS), images (with references), and ground truth (GT) answers for all 1,065 annotated problems in a ZIP file, uploaded as Supplementary Material alongside this paper.

