# OpenReview forum: "SoM-1K: A Thousand-Problem Benchmark Dataset for Strength of Materials"
_ICLR.cc/2026/Conference — Submitted to ICLR 2026_

### Official Review · Reviewer_U3md · 2025-10-20

**Soundness:** 2
**Presentation:** 3
**Contribution:** 2
**Rating:** 4
**Confidence:** 4

**Summary:**

This paper proposes SoM-1K, the first large-scale multimodal benchmark designed to evaluate foundation models on Strength of Materials (SoM) problems that combine textual and schematic information. Experimental results show that even top-performing models achieve only 56.6% accuracy, with LLMs using Descriptions of Images (DoI) outperforming vision-language models—highlighting the need for more advanced multimodal reasoning capabilities in engineering AI.

**Strengths:**

1. Evaluating modern MLLMs on diverse engineering problems is an important and timely contribution.

2. Novelty: SoM-1K is the first domain-specific multimodal benchmark for Strength of Materials.

3. The paper is clearly written and easy to follow.

**Weaknesses:**

1. Human-based evaluation: The evaluation is conducted manually by human annotators rather than automated tools, which may lead to inconsistencies across models or researchers. Furthermore, the need for human evaluators limits the reproducibility and scalability of the benchmark for future studies.

2. Limited model coverage: The paper only evaluates relatively outdated models. More recent models such as o3, Gemini 2.5 Pro, Qwen 2.5-VL series, and InternVL 3 series should be included to provide a stronger and more relevant comparison.

3. Poor figure readability: The figures are difficult to interpret and compare across models and prompts. Presenting the main results in a tabular format would improve readability and facilitate comparison.

4. Weak model baseline in Section 4.3: The analysis and findings in Section 4.3 are based on less capable models. Including results from stronger, state-of-the-art models (e.g., o3, Gemini 2.5 Pro) would make the conclusions more convincing.

5. Questionable novelty of DoI: The proposed Description of Images (DoI) approach appears conceptually similar to prior “describe-then-reason” prompting methods. The key difference—using expert-generated descriptions instead of model-generated ones—limits real-world applicability, since such expert annotations are not available during inference.

6. Unclear reasoning claim: The statement “we propose and validate the use of text-based diagram descriptions (DoI) as an effective prompting strategy to reduce reasoning errors” seems inconsistent. Since DoI is explicitly defined as containing no reasoning (only perceptual content), it likely improves perception accuracy rather than reasoning ability. The paper should clarify this distinction.

**Questions:**

Refer to weaknesses

---

> ### Author Response · Authors · 2025-11-23
> **Response to Reviewer U3md**
>
> We are grateful for the reviewer’s valuable feedback. The reviewer’s concerns have been consolidated, and our point-by-point responses are presented below.
>
> ### 1. **Human-Based Evaluation**
> We agree with the reviewer that the mainstream approach in recent work is to use LLMs-as-Judge to automate the evaluation process. Our pilot study used Qwen-plus to grade 200 sample problems, which achieved an 83% agreement with expert grading. Given our goal of performing detailed error analysis, such agreement is not desirable yet. Hence, we ultimately chose to rely on manual evaluation by human experts for the full dataset. We also concur with the reviewer that manual evaluation may introduce inconsistencies. Hence, we used a simple but rigorous binary grading method, i.e., "1" for correct and "0" for error, to reduce the likelihood of inconsistencies since the final answers for each problem are quantified and fully verifiable.
>
> To address the reviewer's concern, we have mentioned this process in our paper:
>
> *"The evaluation proceeded in two stages: first, …, and second, …. Such binary grading method simplified the process and also reduced the likelihood for inconsistencies."*
>
> Moreover, such datasets, including the model’s responses and our manual grading, can serve as a valuable training set to improve AI evaluators for future automated evaluations.
>
> ### 2. **Concerns about using outdated models and Limited Model Coverage**
>
> To compare the performance of models with different attributes, we selected models with their release dates from 11/2023 to 08/2025 (5 out of 8 models were released after April 2025). It should also be noted that Deepseek-R1 has also been evaluated in other reasoning benchmarks ([Deepseek News](https://api-docs.deepseek.com/news/news250528)) and has demonstrated SOTA performance.
>
> To address the reviewer's concern, we added the latest InternVL3.5 model (released in August 2025) to our evaluation. We tested a random sample of 500 problems from the 1065 dataset, running each for 5 rounds and using the majority vote for manual evaluation. InternVL3.5 achieved 39.8% (PS+I), 52.6% (PS+I+DoI), and 57.0% (PS+DoI). Overall, these results are consistent with our previous findings regarding VLMs' limited ability to interpret domain-specific diagrams and the effectiveness of well-structured textual prompts. We will add the results for internvl3.5-241b-a28b with the entire dataset in the next version of our paper.
>
> | **Model**   | **PS+I** | **PS+I+DoI** | **PS+DoI** |
> |-------------|:--------:|:------------:|:----------:|
> | Internvl3.5-241b-a28b |   39.8%  |    52.6%     |    57.0%   |
>
> We also checked Qwen 2.5-VL, which was superseded by Qwen-VL_Max in our evaluation. Due to budget constraints, Gemini 2.5 Pro and o3 were not included in this work. We thank reviewer's feedback and has mentioned this limitation of our work in the revised paper (line 508).
>
> ### 3. **Poor Figure Readability and Suggesting Tabular Format**
> To address this, we have made several updates in the revised manuscript.
>
> In Appendix C, we provide both a tabular representation (Table 3) and a supplementary figure (Figure 8) of the final results, enabling easier comparison of model performance across different prompts.
>
> ### 4. **Questionable Novelty and Real-world Applicability of DoI**
> Current VLMs have limited capabilities in generating accurate and thorough descriptions of mechanics diagrams needed for effective reasoning. In our experience, more than 80% of generated descriptions contain hallucinations or miss important information. Therefore, expert correction is necessary to ensure the quality and reliability of the DoI annotations.
>
> We also agree with the reviewer that such expert annotations would limit real-world applications. Hence, we are finetuning a Visual Language Model (VLM) framework to automatically generate DoI annotations from mechanics images using the expert-annotated DoI dataset from this work. We mentioned this in the future work as follows:
>
> *"Additionally, the scalability of the DoI annotation process is another challenge, and we are developing a system for automated DoI generation through VLM fine-tuning for a larger dataset."*
>
> ### 5. **Unclear Reasoning Claim**
> We revised the manuscript to clarify that DoI improves the perception accuracy of visual information, reducing reasoning errors during problem-solving.
>
> Revised statement in the paper:
>
> *"We propose and validate the use of Description of the Image (DoI) as an effective prompting strategy to improve the perception capacity of current foundation models."*

---

### Official Review · Reviewer_5Ywi · 2025-11-01

**Soundness:** 2
**Presentation:** 3
**Contribution:** 2
**Rating:** 4
**Confidence:** 3

**Summary:**

This paper introduces SoM-1K, a multimodal benchmark for evaluating foundation models on Strength of Materials (SoM) problems that combine text, equations, and engineering diagrams. The dataset contains 1,065 annotated problems covering axial loading, torsion, and bending, each paired with expert-generated textual diagram descriptions (DoI). The authors benchmark eight large models (LLMs and VLMs) and find that text-based DoI inputs significantly outperform direct visual inputs, highlighting models’ limitations in diagram interpretation. The dataset is well-constructed and addresses a gap in engineering-focused evaluation, though the contribution is primarily dataset-centric with limited methodological novelty.

**Strengths:**

Well-curated and domain-authentic dataset; clear evaluation design isolating visual vs textual reasoning; provides an insightful observation that current VLMs fail on symbolic, structured diagrams while LLMs handle their textualized forms better.

**Weaknesses:**

1. Lack of benchmark validation and answerability calibration.
The paper does not report item difficulty, discrimination, or inter-annotator consistency, making it unclear whether low model scores stem from reasoning limitations or inherently ambiguous or overly difficult problems.
2. Questionable equivalence between visual and textual conditions.
The DoI inputs often encode explicit geometric and boundary relations that are not directly visible to models, making PS+DoI semantically richer than PS+I. This weakens the claim that DoI merely textualizes visual information.
3. Absence of human baselines.
The study relies entirely on expert grading without reporting any human performance or consistency metrics, preventing interpretation of model accuracy (e.g., 56%) relative to realistic human-level baselines.

**Questions:**

1. Have you validated per-item difficulty or answerability (e.g., via item–response analysis or human–model agreement) to ensure benchmark reliability?
2. How do you guarantee that DoI does not introduce extra semantic information beyond the schematic itself? Could you quantify or control for information density differences?
3. What is the human performance level (e.g., engineering students, instructors) on SoM-1K, and how does it compare to model results?

---

> ### Author Response · Authors · 2025-11-23
> **Response to Reviewer 5Ywi**
>
> We thank the reviewer for the constructive comments. Below, we summarize the reviewer’s concerns and present our responses.
>
> ### 1. **Lack of Benchmark Validation and Answerability Calibration**
> The problems in our dataset are sourced from well-recognized textbooks that cover a range of complexities, and they are presented with clear answerability.
>
> To further validate the benchmark, we conducted a human-baseline experiment where four teaching assistants in the discipline of structural engineering solved a randomly selected set of 100 problems (each of them solved 25 problems). The baseline result of 95% accuracy, included in the revised manuscript, offers insights into the difficulty and answerability of the problems (line 342).
>
>
> ### 2. **Clarification on the Equivalence Between Visual and Textual Conditions, and Information Density Control**
>
> To ensure clarity, we generated the DoIs using the following process, which had been added to Appendix B of this paper:
>
> *"The Description of Image (DoI) provides a clear description of the visual elements in engineering schematics. To generate a DoI, follow these steps:"*
>
> *1.Start with the original PNG schematic diagram and process it through the Doubao VLM, which will generate an initial description capturing the geometry (shapes and dimensions of components), loading (applied forces, moments, directions and points of application), and boundary conditions (supports).*
>
> *2.Review and refine the description by correcting errors and adding any missing details from the diagram, such as hidden supports or load directions.*
>
> *3.Perform a final check: the DoI should strictly describe what is visible in the diagram without any assumptions or reasoning—do not infer reactions or deflections that are not shown. Use clear, technical language, organizing the description into sections (Geometry, Loading, Boundary Conditions)."*
>
> Based on our evaluation, the advantage of DoIs stems from their ability to convey the information from the images in a structured and textual format. However, the current VLM's ability to extract such information from the image was shown to be limited. Hence, prompting with PS+DoI could be semantically richer than PS+I, but this is different from chain-of-thought prompting in terms of helping with the model's reasoning ability.
>
> To address the reviewer's concern, we mentioned the difference between the visual and textual inputs for foundation models in the revised paper:
>
> *"We also demonstrated that using DoI as a prompting strategy dramatically improves performance by reducing misinterpretation errors. This suggested that, compared with visual data, well-structured textual input may be semantically richer and could serve as a more reliable foundation for complex reasoning."*
>
> ### 3. **Human Baselines and Benchmark Reliability**
> We agree with the reviewer that the inclusion of human performance baselines is essential for interpreting model accuracy and providing meaningful comparisons between model and human-level reasoning.
>
> To address this, we conducted a baseline experiment with four teaching assistants with a structural engineering background. They were tasked with solving a randomly selected set of 100 SoM problems, with each of them solving 25 problems. They were given two days to solve the problems, with access only to the textbook (including relevant formulas and general knowledge, etc.). The results revealed that the team achieved an accuracy of 95%, while Qwen-plus, with a peak performance of 56.6%, did not reach human-level accuracy.
>
> In the revised version of the paper, we have included the human-baseline results, which offer a clearer comparison between human performance and model performance.
>
> Here is the added statements in Section 4.3:
>
> *"To establish a meaningful performance baseline, we conducted a human evaluation with four teaching assistants from the field of structural engineering, who were tasked with solving a randomly selected set of 100 SoM problems, with each assistant solving 25 problems, within two days, referring only to the textbooks (formulas, general knowledge, etc.). They achieved 95% accuracy, whereas Qwen-plus, with a best accuracy of 56.6%, remains well below human-level performance."*
>
> Furthermore, all the problems in our dataset are sourced from well-recognized textbooks and competition problem sets, which include questions of varying complexity, from easy to more challenging ones.

---

### Official Review · Reviewer_32yG · 2025-11-01

**Soundness:** 2
**Presentation:** 2
**Contribution:** 2
**Rating:** 6
**Confidence:** 2

**Summary:**

The paper SoM-1K: A Thousand-Problem Benchmark for Material Strength presents the first large-scale multimodal dataset designed to evaluate large language and vision-language models for authentic engineering problem solving.  The benchmark includes 1065 textbook-sourced problems covering axial loading, torsion, bending, and composite scenarios, each accompanied by a problem statement, schematic diagram, description of image (DoI), and expert ground truth solution.  Eight cutting-edge models (such as GPT-4o, Claude 3.7, and Qwen-Plus) are tested using various prompt modalities (text + image + DoI).  The results show that even the best model achieves only about 56% accuracy and that textual DoI inputs consistently outperform direct visual inputs, revealing fundamental flaws in current MLLMs' ability to extract and reason from engineering schematics.  The paper presents a rigorously curated dataset, a novel DoI prompting paradigm, and detailed error analysis, establishing SoM-1K as a foundational benchmark for AI-for-mechanics and multimodal scientific reasoning research.

**Strengths:**

- Authentic and Domain-Grounded Benchmarks
The dataset was created directly from real Strength of Materials (SoM) textbooks and competition problems, making it significantly more realistic than synthetic physics or symbolic datasets.  The inclusion of various problem types - axial, torsion, bending, and combined load cases, addresses the key conceptual foundations taught in mechanics courses.

-Well-designed Multimodal Structure
Each item has four components: Problem Statement (PS), Image (I), Description of Image (DoI), and Ground Truth (GT), all of which are tightly aligned and manually verified.  This organization enables the precise identification of where multimodal reasoning fails (visual extraction versus symbolic reasoning).

-Use "Description of Image" (DoI) Input
Introducing DoI as a textual connection between the schematic and reasoning steps is a valuable innovation.  Empirically, DoI prompts significantly improve accuracy when compared to direct visual input, indicating that translating visual information into structured text is still the most effective pathway for current models.

-Comprehensive Model Evaluation
The benchmark compares several leading models (GPT-4o, Claude 3.7, Gemini 2.5, Qwen-VL 2.5, and so on) using three consistent prompt settings (PS + I, PS + DoI, and PS + I + DoI).  This breadth allows for a balanced comparison of text-only, vision-language, and hybrid reasoning approaches.

**Weaknesses:**

-No human or expert baseline.
Without reporting human accuracy, it is difficult to determine whether 56% model performance is near-human or significantly below it.  A reference from engineering students or instructors would help to contextualize the difficulty.

- Evaluation Metric
The binary correctness score does not account for partial reasoning quality, such as correct free-body setup versus numerical slip. A step-by-step or partial-credit evaluation could be more effective in measuring reasoning depth.

- Scalability of DOI Annotation
The DoI layer currently relies on manual verification, which may limit its scalability to larger or cross-domain datasets. The paper could have gone into greater detail about how to automate or standardize this process.

**Questions:**

Annotation Quality - How many annotators verified the DoI descriptions, and how was inter-annotator consistency assessed?

Human Baseline - Have you tested the SoM-1K with experts to determine typical human accuracy?

DoI Generation - Were the DoIs generated by a machine and then edited, or were they entirely written by humans?  Could future datasets rely on automated generation?

Practical Impact - Beyond benchmarking, how do you see SoM-1K being used? For model fine-tuning, educational feedback, or AI-aided design?

Prompt Fairness - Were all models evaluated using the same context length, temperature, and image resolution to ensure an accurate comparison?

---

> ### Author Response · Authors · 2025-11-23
> **Response to Reviewer 32yG**
>
> We sincerely appreciate the reviewer’s thoughtful comments. A summary of the reviewer’s concerns and our corresponding responses is provided below.
>
> ### 1. **Absence of Human or Expert Baseline**
> To address this, we conducted a baseline experiment with four teaching assistants from the field of structural engineering. Each assistant solved 25 randomly selected SoM problems, totaling 100 problems, over the course of two days referring only the textbooks (formulas, general knowledge, etc.). The team achieved 95% accuracy, which is much higher than the best performing model, i.e., Qwen-plus of 56.6%.
>
> We have included these human baseline results in the revised version of the paper to offer a more meaningful comparison between human and model performance (line 342).
>
> ### 2. **Limitations of the Binary Scoring Method**
> The authors appreciated the comments from the reviewer. In the current practice of structural engineering, engineers focus more on the correctness of the final answer to the mechanics problem. Hence, we followed a strict binary correctness method to evaluate the capacity of these foundation models in the full 1065 mechanics questions.
>
> It should also be noted that we further performed error type analysis on the 100 problems where all models failed. During the analysis, our evaluators checked the detailed reasoning process of each model, and pointed out the detailed error types.
>
> We agree with the reviewer that partial-credit evaluation could serve as a more direct method in measuring reasoning depth of LLMs/VLMs. Due to the limited time and resources available, we added this limitation in our revised manuscript as follows:
>
> *"Additionally, our current binary correctness score does not capture partial reasoning, such as correct free-body setups with numerical errors, and a more granular step-by-step or partial-credit evaluation would provide deeper insights into model reasoning."*
>
> ### 3. **Scalability of DoI Annotation**
> At the beginning of this project, we used VLMs to generate DoIs of images for each problem. However, we found that the automated DoIs from VLMs could miss/misinterpret critical information from the image, and human correction was therefore needed to further evaluate the models’ reasoning capacity. We agree with the reviewer that the current reliance on manual correction for the DoI layer could limit its scalability.
>
> To address this, we are currently developing a Visual Language Model (VLM) framework designed to automatically generate DoI annotations from images (through supervised fine-tuning) using our manually annotated DoIs. We believe the current DoI annotation is necessary and our ongoing work will help reduce the manual workload and improve the efficiency of the annotation process.
>
> In the revised paper, we mentioned the rationale for including human correction in the DoI generation pipeline and its limitations (line 512).
>
> ### 4. **Question about Annotation Quality**
> Our annotation team consists of experts in structural engineering and mechanics of materials, including a PhD candidate, two lecturers, and four teaching assistants. To ensure the quality of the DoI, multiple annotators (at least two) reviewed the content, and discrepancies were resolved through regular discussions and a consensus-based approach.
>
> To address the reviewers' concern, we provided our guidelines for correcting DoIs in Appendix B:
>
> *"The Description of Image (DoI) provides a clear description of the visual elements in engineering schematics. To generate a DoI, follow these steps:..."*
>
>
>
> ### 5. **Question about Practical Impact**
> Beyond benchmarking, we envision SoM-1K being used for model fine-tuning in the domain of Strength of Materials (SoM) and for educational applications, helping students improve their problem-solving skills with real-time feedback. Moreover, SoM-1K can serve as a valuable resource for advancing Vision-Language Models (VLMs) toward stronger mechanics-engineering capabilities, where current VLMs continue to face significant challenges.
>
> In the revised paper, we also mentioned this in the end:
>
> *"SoM-1K could also be leveraged for practical applications, such as model fine-tuning for specific engineering tasks and AI tutoring to help students improve problem-solving skills and understand complex concepts."*
>
> ### 6. **Question about Prompt Fairness**
>
> We used the same prompt settings for different models for fair comparisons. In the revised paper, we added this statement in Section 4.2:
>
> *"Additionally, we did not impose any token length restrictions or modify parameters such as temperature, and all settings were used with their default values, ensuring that the model could complete the full reasoning process and output complete results."*

---

### Official Review · Reviewer_DEt5 · 2025-11-03

**Soundness:** 2
**Presentation:** 2
**Contribution:** 2
**Rating:** 4
**Confidence:** 4

**Summary:**

The paper introduces SoM-1K, a 1,065-problem multimodal benchmark for strength of materials that pairs textual statements with schematic diagrams and descriptions of images (DoI) to textualize visuals. It evaluates multiple LLM/VLM models under three prompting regimes with majority voting over five generations and expert binary grading.

**Strengths:**

- Evaluation uses multiple prompts per item, majority vote, and human grading
- Results show that DoI reduces visual misinterpretation (extraction) errors across VLMs and current models are more reliable with precise text than raw schematics.

**Weaknesses:**

- The paper fails to cite or compare against existing multimodal benchmarks such as MMMU [1], which is broader and more comprehensive, covering multiple academic subjects and modalities. Without such comparison, the novelty and relative contribution of SoM-1K remain unclear. The authors should explicitly position their work within the broader landscape of multimodal reasoning benchmarks, clarifying what unique capabilities or insights it offers beyond existing large-scale datasets.

- The paper does not include a human performance baseline (e.g., results from students or experts solving the same problems). The absence of such a reference makes it difficult to assess how large the gap remains between current models and expert-level reasoning.

- Figure 4 is hard to interpret as all models, prompting strategies, and open/closed-source systems are plotted together, making trends difficult to distinguish. Separating results by model type (open vs. closed) and prompting strategy would improve clarity and make performance comparisons more meaningful.

- Relying solely on accuracy may mask variation in partial-credit reasoning quality. The paper would benefit from per-subcategory breakdowns (e.g., determinate vs. indeterminate, frames vs. beams).

- The evaluation of VLMs lacks sufficient details regarding image preprocessing and resolution settings, which can critically affect visual reasoning performance. The authors should specify how diagrams were rendered, scaled, and fed into each model. Moreover, it would strengthen the study to explore whether structured diagram parsing (e.g., converting schematic images into vectorized primitives or symbolic graph representations) can reduce the performance gap between the PS+I (image-based) and PS+DoI (textualized diagram) conditions. Such analysis would help clarify whether the observed advantage of DoI stems from richer semantics or simply from preprocessing limitations in visual input.


[1] Yue, Xiang, et al. "Mmmu: A massive multi-discipline multimodal understanding and reasoning benchmark for expert agi." Proceedings of the IEEE/CVF Conference on Computer Vision and Pattern Recognition. 2024.

**Questions:**

see weaknesses above

---

> ### Author Response · Authors · 2025-11-23
> **Response to Reviewer DEt5**
>
> The authors appreciate the reviewer's insightful comments. We have summarized the reviewer’s concerns and provided our detailed responses below.
>
> ### 1. **Lack of Comparison with Existing Multimodal Benchmarks**
>
> In the revised version of the paper, we acknowledged the contributions of the MMMU paper and explicitly positioned SoM-1K within the broader context of multimodal reasoning benchmarks.
>
> We added the following statements:
>
> *"More recently, Yue et al. (2024) introduced the MMMU benchmark to evaluate the multimodal reasoning capabilities of various models across a broad range of disciplines. To the best of the authors’ knowledge, however, benchmark studies specifically designed for mechanics analysis, particularly those with detailed reasoning steps as ground truth and explicit error-type classifications, remain scarce."*
>
> ### 2. **Absence of Human Performance Baseline**
> We agree with the reviewer that a human performance baseline is needed for comparison.
>
> To address this, we randomly selected a set of 100 problems and asked four teaching assistants with backgrounds in structural engineering to solve these problems within two days (each of them solved 25 problems). They could only refer to the textbooks (including formulas, general knowledge, etc.). The final results showed that the expert team achieved an accuracy of 95%, whereas Qwen-plus, with its best performance at 56.6%, falls below human-level accuracy.
>
> In the revised version of the paper, we have included the human baseline results:
>
> *"To establish a meaningful performance baseline, we conducted a human evaluation with four teaching assistants from the field of structural engineering, who were tasked with solving a randomly selected set of 100 SoM problems, with each assistant solving 25 problems, within two days, referring only to the textbooks (formulas, general knowledge, etc.). They achieved 95% accuracy, whereas Qwen-plus, with a best accuracy of 56.6%, remains well below human-level performance."*
>
> ### 3. **Confusion in Figure 4**
> The original figure was to showcase the overall ranking of the performance from different models and prompting strategies.
>
> To improve clarity, we have added a new table (Table 3) and figure (Figure 8) in Appendix C (due to page length limitations) to showcase the results by model type (open vs. closed-source) and prompting strategy. This will allow for a more focused comparison and help highlight the differences in performance across different configurations, making the trends easier to interpret and more informative.
>
> ### 4. **Per-Subcategory Breakdown**
> We agree with the reviewer and have added figures (Figure 9) in Appendix C that break down the performance of each model by subcategory, such as determinate vs. indeterminate structures and different loading types (axial loading, torsion, bending, integrated tasks, etc.).
>
> In the revised version of the paper, we have presented new insights regarding category-specific performance in Section 4.3 (line 327 and line 339).
>
> ### 5. **Lack of Details on Image Preprocessing and Resolution**
> In the revised manuscript, we have clarified that the diagrams were manually scanned from textbooks, uploaded to the cloud, and fed into the models via URLs without other traditional image processing. In particular, no rendering or scaling was applied to the images, which simulated a school setting where students can take a picture of the problem and feed it to a model to get the final answer. In our dataset, the range of image resolutions is 195x104 to 850x688, which was added to the paper for readers' reference:
>
> *"First, schematic diagrams were manually extracted and stored as PNG images (size ranging from 195x104 to 850x688), uploaded to the cloud, and fed into models via URL links."*
>
>
> ### 6. **Effects of Structured Diagram Parsing on Visual Reasoning Performance**
>
> Thanks for this valuable feedback. We agree with the reviewer that converting schematic images into vectorized primitives or symbolic graph representations is crucial for further uncovering the underlying mechanisms behind PS+I and PS+DoI prompting. However, due to time and resource constraints, we acknowledged this as a limitation in the revised paper:
>
> *"Several limitations should be addressed in future work. One key area is the impact of image rendering and diagram parsing on the reasoning performance of VLMs."*

---

### Author Response · Authors · 2025-12-03

Dear Program Chairs, Senior Area Chairs, and Area Chairs,

We thank the reviewers for their constructive feedback. **32yG** and **5Ywi** commended the **authenticity and domain-grounded nature** of SoM-1K, emphasizing its value as a realistic benchmark derived from standard textbooks. **DEt5** and **32yG** highlighted the **principled design of our evaluation framework** (specifically the inclusion of DoI), noting its effectiveness in isolating visual extraction failures from symbolic reasoning errors; **U3md** recognized the **novelty and timeliness** of the work as the first specialized multimodal benchmark for Strength of Materials. During the revision stage, we have made substantial updates: (i) **conducted and reported a comprehensive human performance baseline experiment** to validate benchmark difficulty and answerability; (ii) **extended our evaluation to the latest InternVL3.5 model on 500 randomly sampled SoM-1K problems**; (iii) **added detailed breakdown figures and tables** in **Appendix C** to improve readability and allow for granular analysis; (iv) **clarified DoI annotation guidelines**; (v) **explicitly positioned our work against general benchmarks like MMMU**; (vi) **provided additional details on image preprocessing and resolution**; and (vii) **expanded the "Limitations" and "Future Work" sections** to rigorously address partial scoring, **Description of Image (DoI)** automation, and VLM diagram parsing.

We respectfully summarize our detailed responses below:

**1. Addressed Concerns on Human Baselines and Benchmark Validity (DEt5, 32yG, 5Ywi)**

**[Human Performance Baseline]:** Addressing concerns about benchmark difficulty, we conducted a controlled experiment with four structural engineering teaching assistants. They solved randomly selected 100 problems (25 each) over 2 days with access only to standard textbooks (formulas, general knowledge, etc.).
*   **Result & Comparison:** The expert team achieved **95% accuracy**, contrasting sharply with the best model (Qwen-plus) at **56.6%**.
*   **Implication:** This gap confirms SoM-1K is solvable by human experts yet remains challenging for SOTA models, validating problem answerability while highlighting the reasoning gap. We integrated these findings into Section 4.3 to provide context for model performance.

**2. Clarified DoI Methodology: Scalability, Quality, and Purpose (32yG, 5Ywi, U3md)**

**[Scalability & Automation]:** We clarified that manual DoI correction is currently necessary because VLMs suffer high hallucination rates (~80%) on complex schematics. We added to **Future Work** that this dataset is intended to serve as a high-quality training corpus to fine-tune future VLMs for automated DoI generation, thereby solving the scalability issue in the long term.

**[Perception vs. Reasoning]:** We addressed the query regarding whether DoI aids reasoning or just perception. We clarified that DoI primarily bridges the **perception gap** (visual-to-text). By isolating the visual perception failure, we demonstrated that models perform significantly better given correct textual descriptions, proving that the bottleneck often lies in extracting visual constraints rather than purely in symbolic reasoning.

**[Annotation Guidelines]:** For reproducibility, we added explicit annotation guidelines to **Appendix B**.

**3. Expanded Model Coverage and Evaluation Metrics (32yG, U3md)**

**[New Model: InternVL3.5]:** Addressing model coverage (U3md), we evaluated the new **InternVL3.5** (released after Aug 2025) during rebuttal on a randomly sampled subset of 500 problems from SoM-1K. It achieved **39.8% (PS+I), 52.6% (PS+I+DoI), and 57.0% (PS+DoI)**, still far below human experts (95%), highlighting the substantial performance gap that persists. These results are also consistent with our previous findings regarding **VLMs’ limited ability to interpret domain-specific diagrams** and the **effectiveness of well-structured textual prompts**. Full results will be included in the next version of the paper.

**[Binary vs. Partial Scoring]:** Regarding partial credits (32yG), we adopt **strict binary scoring** because engineering practice prioritizes the correctness of the final answer for safety; while **partial-credit evaluation** could reveal reasoning depth, it is beyond our current scope and is included as a limitation for future work.

**4. Improved Presentation and Contextualization (DEt5, U3md)**

Addressing **Figure 4**'s readability, we **added Table 3** and **Figure 8** in **Appendix C** to provide a clear tabular comparison of performance by model type (open vs. closed-source) and prompting strategy. We also added **Figure 9** for subcategory breakdowns (e.g., determinate vs. indeterminate structures).

We believe these revisions addressed the core concerns raised by the reviewers and significantly strengthened our paper.

---

### Meta-Review · Area_Chair_qGnT · 2026-01-06

**Summary:**

This work introduce a new multi-modal benchmark in the strength of materials (SoM). The data set contains 1,065 SoM problems, where each one has textual problem statements,  schematic diagrams, descriptions of images (DoI) and ground truth. 8 LLMs and VLMs are evaluated on the benchmark, where the best-performing model achieving only 56.6\% accuracy. The performance difference between LLMs and VLMs is also investigated according to the usage of DoI.

**Reviewer Concerns:**

Before rebuttal, all reviewers had the concern about lacking of human performance on the proposed benchmark. In addition, the details of the evaluation were also missing/insufficient as indicated by Reviewer DEt5, 5Ywi and U3md. Reviewer 32yG, 5Ywi and U3md also had the concern about the DoI annotation. The rebuttal provided a preliminary result from a team of human while the result from a single person may be more convincing. More details about the benchmark were provided in the rebuttal that helped mitigate the concerns about the evaluation settings. For DoI annotations, the rebuttal confirmed that they are reviewed and corrected manually. The process makes the scalability challenging and the comparison between LLM+DoI and VLM+I unfair.

**Reviewer Scores:**

The submission received 4,6,4,4 before rebuttal. Since the rebuttal did not address all major concerns, the scores may not be increased after the discussion.

---

### Decision · Program_Chairs · 2026-01-26

Reject